# Multiscale characterization reveals oligomerization dependent phase separation of primer-independent RNA polymerase nsp8 from SARS-CoV-2

Jinxin Xu[1,2,9], Xin Jiang[3,9], Yulong Zhang[1,2], Yan Dong[1], Changli Ma[4], Hanqiu Jiang [4], Taisen Zuo[4], Rui Chen[1,5], Yubin Ke[4], He Cheng[4], Howard Wang [3✉] & Jinsong Liu [1,2,6,7,8✉]

RNA replication and transcription machinery is an important drug target for fighting against coronavirus. Non-structure protein nsp8 was proposed harboring primase activity. However, the RNA primer synthesis mechanism of nsp8 is still largely unknown. Here, we purified dimer and tetramer forms of SARS-CoV-2 nsp8. Combined with dynamic light scattering, small-angle neutron scattering and thermo-stability analysis, we found that both dimer and tetramer become loosened and destabilized with decreasing salt concentration, and the dimer form is more stable than the tetramer form. Further investigation showed that nsp8 dimer and tetramer can undergo phase separation but exhibit different phase separation behaviors. Nsp8 dimer can form liquid-like droplets in the buffer with a low concentration of NaCl; phase separation of nsp8 tetramer depends on the assistance of RNA. Our findings on different phase separation behaviors of nsp8 dimer and tetramer may provide insight into the functional studies of nsp8 in coronavirus.

[1] State Key Laboratory of Respiratory Disease, Guangzhou Institutes of Biomedicine and Health, Chinese Academy of Sciences, Guangzhou 510530, China. [2] Graduate University of Chinese Academy of Sciences, Beijing 100049, China. [3] Neutron Science Platform, Songshan Lake Materials Laboratory, Dongguan, Guangdong 523808, China. [4] Spallation Neutron Source Science Center, Dongguan, Guangdong 523803, China. [5] School of Life Sciences, University of Science and Technology of China, Hefei 230026, China. [6] Guangdong Provincial Key Laboratory of Biocomputing, Guangzhou Institutes of Biomedicine and Health, Chinese Academy of Sciences, Guangzhou 510530, China. [7] China-New Zealand Joint Laboratory on Biomedicine and Health, Guangzhou Institutes of Biomedicine and Health, Chinese Academy of Sciences, Guangzhou 510530, China. [8] Guangdong-Hong Kong-Macao Joint Laboratory of Respiratory Infectious Diseases, Guangzhou Institutes of Biomedicine and Health, Chinese Academy of Sciences, Guangzhou 510530, China. [9]These authors contributed equally: Jinxin Xu, Xin Jiang. ✉email: wangh@sslab.org.cn; liu_jinsong@gibh.ac.cn

nfection of coronaviruses, a group of positive-strand RNA viruses, is a great threat to human health. The current pandemic coronavirus disease 2019 (COVID-19) is caused by a new coronavirus SARS-CoV-2[1]. Despite extensive applications of vaccines, COVID-19 remains a threat to the global economy and health, due to the shortage of effective therapeutic agents against SARS-CoV-2. Considering the essential role in the life cycle of RNA viruses and the lack of homolog in host cells, RNA replication, and transcription machinery is an attractive therapeutic target against SARS-CoV-2[2,3]. At least two RNA-dependent RNA polymerases (RdRp) are encoded by the coronavirus genome, including primer-dependent RdRp nsp12[4] and primer-independent RdRp nsp8[5,6]. Recent structural studies on SARS-CoV nsp12[7] and SARS-CoV-2 nsp12[8,9] have characterized the RNA elongation mechanism of nsp12. It was proposed that nsp8 is capable of de novo RNA synthesis with an ssRNA template and then providing primers required for nsp12[5]. Although structures of nsp8 from several coronaviruses were reported, the mechanism of nsp8 de novo RNA synthesis is still largely unknown.

It had been reported that, for a number of viruses, the replication and assembly were taken place within granular structures termed "viral factories"[10]. Recently, the viral factory was proposed to be a membrane-less organelle driven by liquid-liquid phase separation (LLPS)[11]. Concentrating reactive molecules in condensates via liquid-liquid phase separation was considered a powerful mechanism to accelerate biochemical reactions[11–14]. Proteins tended to LLPS were proposed to exhibit features including intrinsically disordered, modularity, nucleic acid binding, and oligomeric nature, et al.[11,15]. Sequence analysis showed that SARS-CoV-2 nsp8 contains a potential IDR (intrinsically disordered region) at the N terminus (Supplementary Fig. 1) critical for RNA binding[16] and a conserved catalytic D/ExD/E motif as characterized in SARS-CoV nsp8[6]. Structural analysis also showed that the free N terminus is flexible in several reported structures[16,17], implying that nsp8 may undergo phase separation. Here, we investigated the LLPS of nsp8 of SARS-CoV-2, unveiling oligomerization-dependent phase separation behavior of nsp8.

## Results

### SARS-CoV-2 nsp8 forms dimer and tetramer in solution.
We expressed and purified SARS-CoV-2 nsp8 with a C terminal 8xHis-tag. During the purification process, we found that nsp8 forms oligomers and nucleic acid contamination when the purification buffer contains 300 mM NaCl. To remove the nucleic acids contamination, the concentration of NaCl in the purification buffer was increased to 1 M. Interestingly, nsp8 was eluted from the gel filtration column as two peaks (Fig. 1a). The oligomerization state of nsp8 from two elution peaks was analyzed using dynamic light scattering (DLS) and small-angle neutron scattering (SANS). The calculated molecular weight of nsp8 from the first and second elution peak was 90–93 kDa and 40–49 kDa, respectively (Fig. 1b). As the theoretical molecular weight of nsp8-His is 22.98 kDa, our data indicated that, in addition to the dimer of nsp8 as reported previously[17], a tetramer form of nsp8 was identified. To test whether the dimer and tetramer of nsp8 were in dynamic equilibrium, here, the dimer or tetramer of nsp8 eluted from gel filtration was collected and analyzed by gel filtration again after standing for 10 days. Gel filtration analysis showed that oligomer state changes could be observed for only a small portion of dimer or tetramer (Supplementary Fig. 2), suggesting the oligomer state of nsp8 is rather stable in solution. Because nsp8 contains several cysteine residues, it is worthwhile to investigate whether nsp8 dimer or tetramer formation depends on the disulfide bond. To clarify this, dimer and tetramer form

nsp8 were analyzed by SDS-PAGE with or without reducing agent β-mercaptoethanol. SDS-PAGE results showed that both dimer and tetramer form nsp8 almost completely turned to monomer by SDS (Supplementary Fig. 3), even without reducing agent, indicating that oligomerization of nsp8 is independent of the disulfide bond.

Then, we characterized the shape of the nsp8 dimer and tetramer based on the DLS and SANS measurements. Hydrodynamic radius ($R_h$) as measured by DLS reflects the solvated protein size, while the radius of gyration ($R_g$) as measured by SANS reflects mostly the compositional distribution of hydrogenated protein molecules. The ratio of $R_g$ to $R_h$ ($R_g/R_h$) offers the shape information of protein molecules[18]. The larger $R_g/R_h$ values of nsp8 dimers than those of nsp8 tetramers imply the variation of the shape of molecules from non-spherical to globular (Fig. 1c, d).

### Destabilization of both nsp8 dimer and tetramer as salt concentration decreases.
Interestingly, DLS analysis demonstrated that the $R_h$ of nsp8 dimer and tetramer increased with decreasing NaCl concentration (Fig. 1c). This corroborated well with the SANS measurements that revealed the increase of the $R_g$ at lower NaCl concentrations for both nsp8 dimer and tetramer (Fig. 1d, e). Both $R_h$ and $R_g$ of dimers and tetramers increase noticeably when the concentration of NaCl is decreased, indicating a looser structure in the buffer containing a low concentration of NaCl. These data may suggest that lowering salt concentration destabilizes the structure of both nsp8 dimers and tetramers. To test this hypothesis, we assessed the thermo-stability of nsp8 dimer and tetramer, and showed that both the melting temperature ($T_m$) and the onset temperature of aggregation ($T_{agg}$) indeed decreased at lower NaCl concentration for both nsp8 dimer and tetramer (Fig. 1f, g). Furthermore, a quantitative comparison of $T_m$ and $T_{agg}$ values of the two forms of nsp8 indicates that the dimer form is more stable than the tetramer one.

### Different phase separation behavior of nsp8 dimer and tetramer.
The expanded size and decreased stability with decreasing NaCl concentration imply that nsp8 dimer and tetramer may undergo phase separation at low NaCl concentration. To test this, we directly diluted nsp8 with a low NaCl concentration buffer. Liquid-like droplets appeared in 1 mg/mL nsp8 dimer solution containing 50 mM NaCl (Fig. 2a). When the concentration of NaCl was increased to 100 mM, nsp8 dimer failed to undergo phase separation at 1 mg/mL (Fig. 2a), whereas at 2 mg/mL, nsp8 dimer could form liquid-like droplet in the buffer containing 100 mM NaCl (Fig. 2b). For fluorescent microscopy observation, nsp8 was labeled with His-tag labeling dye RED-tris-NTA. Fluorescent microscopy confirmed that nsp8 dimers formed liquid-like droplets at protein concentration of 1 mg/mL in 50 mM NaCl buffer, and nsp8 condensates could be dissolved upon increasing the NaCl concentration to 275 mM by mixing nsp8 dimer (1 mg/mL) dissolved in 500 mM NaCl with volume ratio 1:1 (Fig. 2c). The liquid-like nature of the droplet was further confirmed by fluorescence recovery after photobleaching (FRAP) experiment as a rapid recovery of fluorescence was observed after photobleaching (Fig. 2d). These results indicate that nsp8 dimer undergoes LLPS in a protein and NaCl concentration-dependent manner.

The thermal stability analysis clearly showed that the nsp8 tetramer is less stable than the dimer (Fig. 1e, f). Then, we tested whether nsp8 tetramers exhibit different phase separation behaviors compared to the dimer. Consistent with our hypothesis, nsp8 tetramer formed solid-like sediments instead of a liquid-like droplet at a concentration of 1 mg/mL in the buffer containing 50

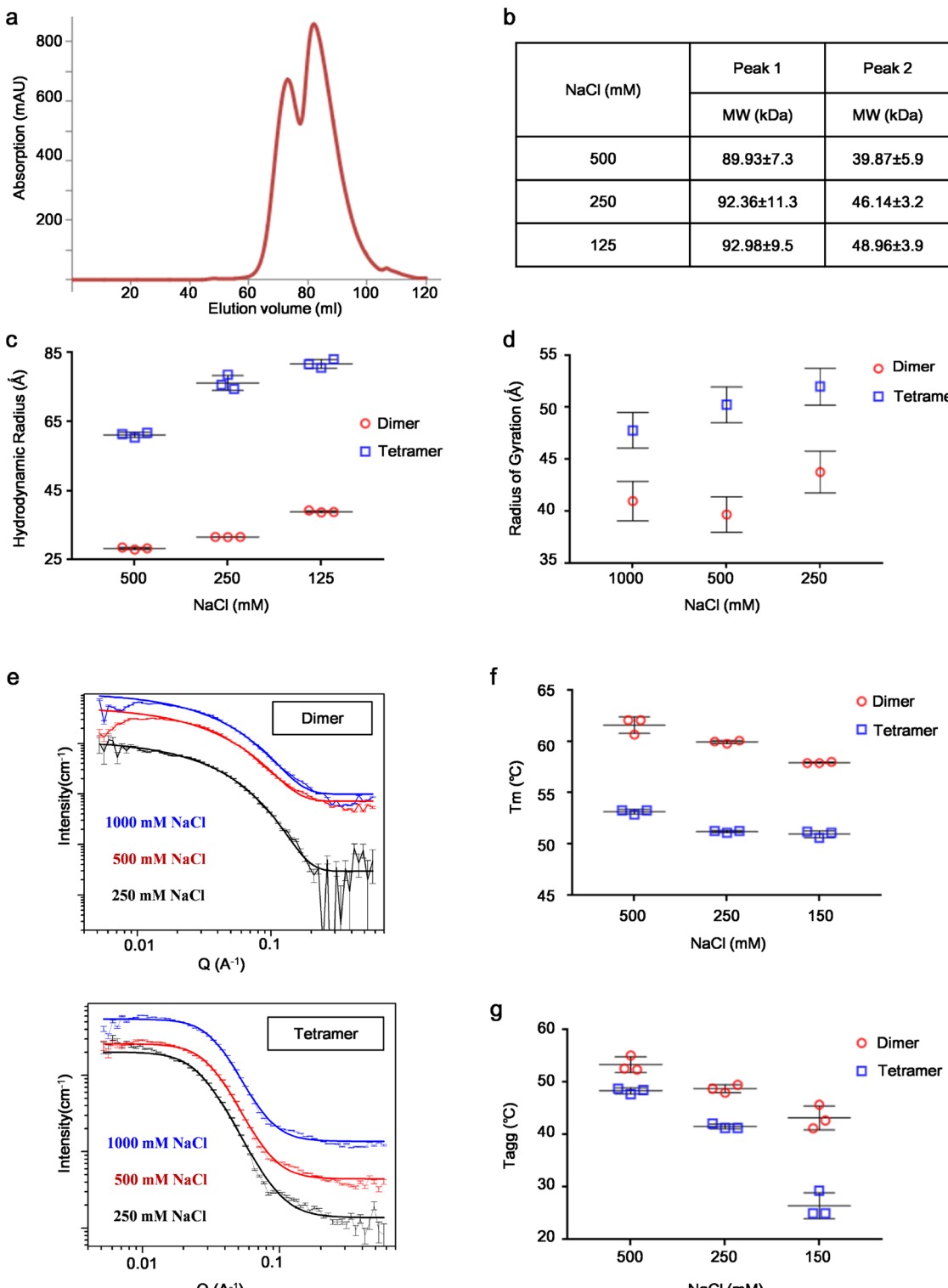

**Fig. 1 Characterization of SARS-CoV-2 nsp8. a** Gel filtration analysis of nsp8. **b** Molecular weight determination of nsp8 from two eluted peaks using DLS. **c** Hydrodynamic radius of nsp8 dimers and tetramers determined by DLS. Data measured by DLS are representative of three independent experiments. Error bars correspond to standard deviation. Protein was dissolved in a buffer of 20 mM HEPES pH 7.4, NaCl (125, 250, or 500 mM). **d** Radius of gyration ($R_g$) of nsp8 dimers and tetramers in buffers containing different concentrations of NaCl. The $R_g$ quantities were obtained from model fitting to the SANS spectra of corresponding solutions as shown in **e**, error bars correspond to SANS fitting error. **f** Melting and **g** aggregation temperature of nsp8 dimers and tetramers. Data were representative of three independent experiments. Error bars correspond to standard deviation. Protein was dissolved in a buffer of 20 mM HEPES pH 7.4 and NaCl (150, 250, or 500 mM). Source data for (**b**–**d**, **f**, **g**) are available in Supplementary Data 1.

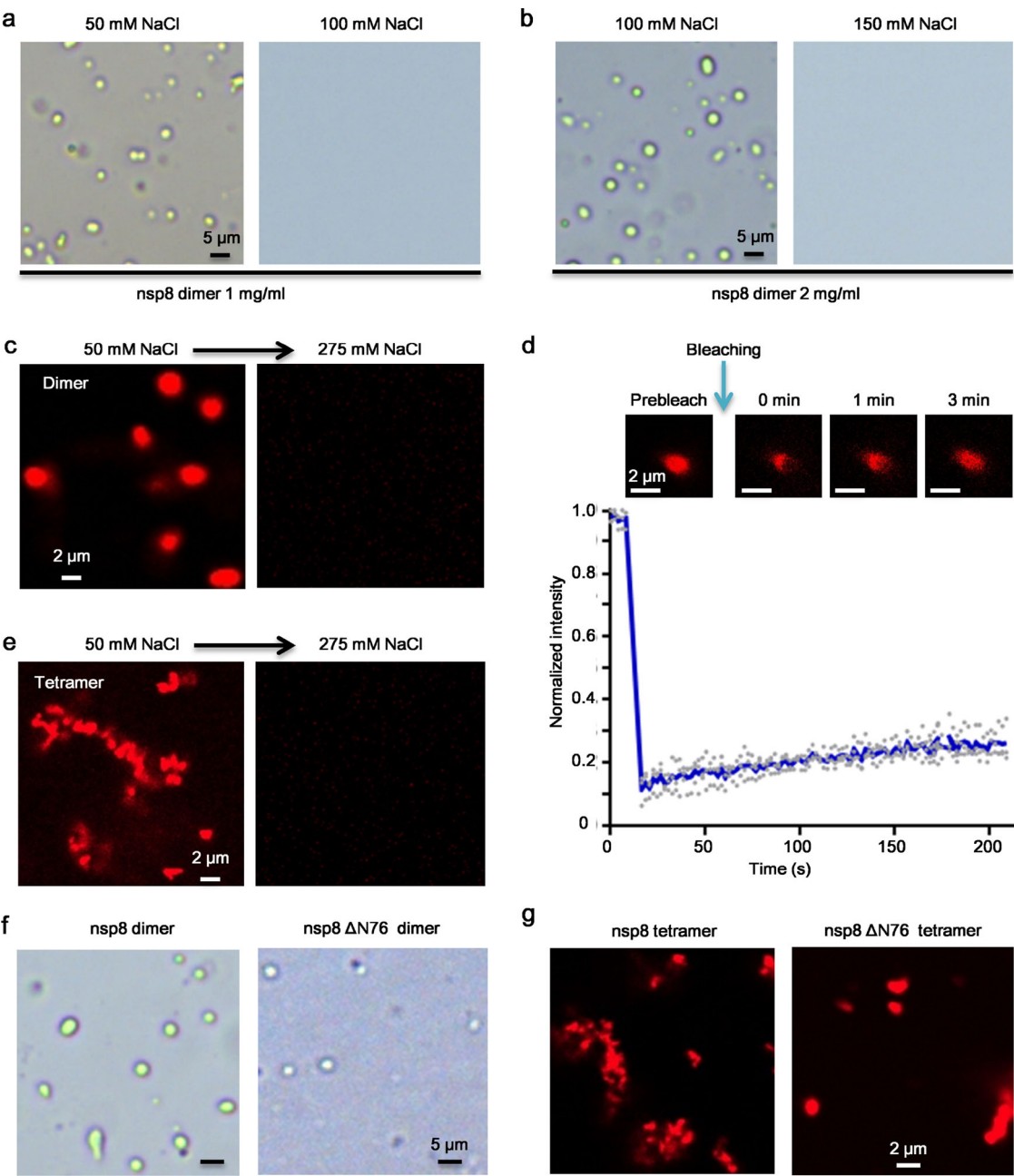

**Fig. 2 Phase separation of SARS-CoV-2 nsp8. a** LLPS assay of nsp8 dimer at 1 mg/mL in the buffer of 20 mM HEPES pH 7.4, 50 or 100 mM NaCl. **b** LLPS assay of nsp8 dimer at 2 mg/mL in the buffer of 20 mM HEPES pH 7.4, 100 or 150 mM NaCl. **c** Liquid-like droplets formed by RED-tris-NTA labeled nsp8 dimers at 1 mg/mL in the buffer of 20 mM HEPES pH 7.4, 50 mM NaCl, can be dissolved upon increasing NaCl concentration to 275 mM by mixing nsp8 dimer (1 mg/mL) dissolved in 500 mM NaCl with volume ratio 1:1. **d** In vitro FRAP analysis of the condensates formed by RED-tris-NTA labeled nsp8 at 1 mg/mL in a buffer of 20 mM HEPES pH 7.4, 50 mM NaCl. Data were representative of three independent experiments; error bars represent standard deviation. Source data is available in Supplementary Data 1. **e** nsp8 tetramer at 0.25 mg/mL forms solid-like condensates in the buffer of 20 mM HEPES pH 7.4, 50 mM NaCl. The solid-like condensates can be dissolved upon increasing NaCl concentration. **f** Deletion of N-terminal 76 residues inhibits the phase separation of nsp8 dimer in a buffer of 20 mM HEPES pH 7.4, 100 mM NaCl. The concentration of wildtype or mutated nsp8 dimer was 2 mg/mL. **g** Deletion of N-terminal 76 residues reduces the aggregation of nsp8 tetramer in a buffer of 20 mM HEPES pH 7.4, 50 mM NaCl. The concentration of wildtype or mutated nsp8 tetramer was 0.25 mg/mL.

or 100 mM NaCl (Supplementary Fig. 4). Solid-like condensates could still be observed even as the concentration of nsp8 tetramer decreased to 0.25 mg/mL with low NaCl concentration, which can be reversed by simply increasing NaCl concentration to 275 mM (Fig. 2e).

It was reported that nsp7 could increase the primer extension activity of nsp8 via forming a complex with nsp8 in SARS-CoV[6].

Hence, we test if the phase separation of nsp8 can be enhanced or disrupted by nsp7. As shown in Supplementary Fig. 5, when mixing with nsp7, the phase transformation behavior of the nsp8 dimer or nsp8 tetramer did not exhibit obvious changes.

To investigate whether the N-terminal IDR is essential for phase transformation, we generated a truncation of nsp8 by deleting the N-terminal 76 residues (nsp8ΔN76). Similar to the

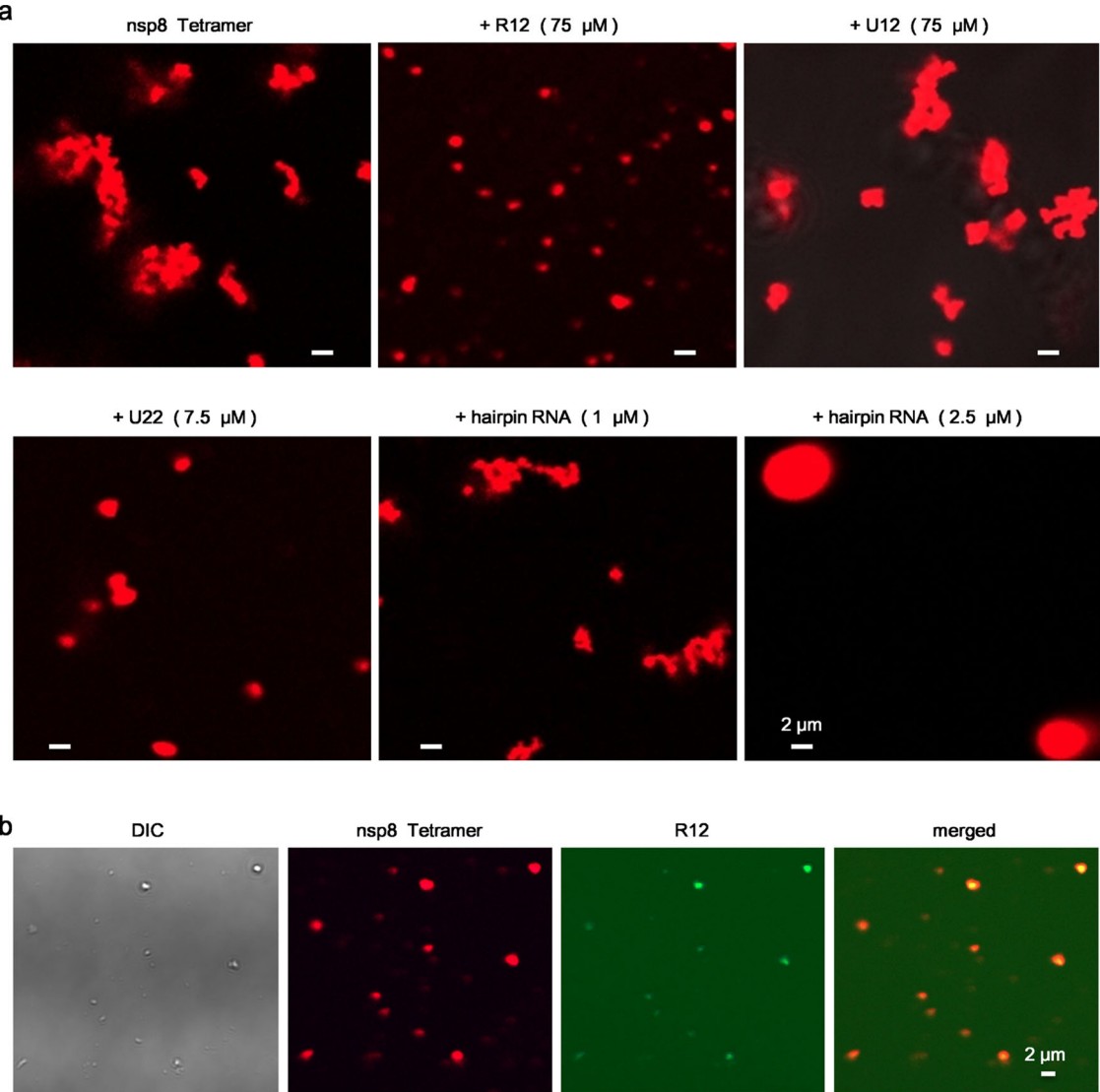

**Fig. 3 RNA modulates phase separation behaviors of nsp8 tetramer. a** RNA induces LLPS of nsp8 tetramer in a sequence and length-dependent manner. Before imaging, nsp8 tetramer at 0.25 mg/mL was labeled with RED-tris-NTA and mixed with RNA in a buffer of 20 mM HEPES pH 7.4, 50 mM NaCl. **b** Co-localization of nsp8 tetramer and ssRNA R12 in liquid-like droplets. Before imaging, 0.25 mg/mL RED-tris-NTA labeled nsp8 tetramer was mixed with 75 μM 6-FAM labeled R12 in a buffer of 20 mM HEPES pH 7.4, 50 mM NaCl.

wild-type nsp8, nsp8ΔN76 also formed homodimer and tetramer in solution (Supplementary Fig. 6). However, the deletion of N-terminal 76 residues partially abolished the droplet formation of nsp8 dimer (Fig. 2f), suggesting that the N-terminal IDR involves in LLPS of nsp8 dimer. Interestingly, with the deletion of N-terminal 76 residues, the truncated nsp8 tetramer was less aggregated at 50 mM NaCl compared to the wild-type nsp8 tetramer (Fig. 2g). This observation suggested that the N-terminal IDR plays a key role in aggregation of nsp8 tetramers at decreasing NaCl concentration.

**RNA modulates LLPS of nsp8 tetramer**. The N terminus of nsp8 is a defined RNA binding motif, which can be stabilized by binding with RNA[8,16]. It was proposed that the droplet-forming property of protein might be altered after binding to RNA[15]. Hence, we hypothesized that RNA binding may induce nsp8 tetramer transition from solid-like condensate to LLPS at low NaCl concentration. To confirm this, we designed a 12-nt ssRNA (R12) with a sequence derived from the SARS-CoV-2 genome adjacent to poly (A). When mixing RED-tris-NTA labeled nsp8

tetramer (0.25 mg/mL) with R12, a liquid-like droplet could be observed with RNA concentration up to 75 μM (Fig. 3a). To verify whether RNA modulating LLPS of nsp8 tetramer depends on the sequence and the length, LLPS experiments were performed in the presence of 12-nt poly U (U12) or 22-nt poly U (U22) (Fig. 3a). We found that, when mixed with 75 μM U12, nsp8 tetramer still formed solid-like structure instead of a liquid-like droplet. However, the nsp8 tetramer turned to a liquid-like droplet when mixing with 22-nt poly U (U22) at concentrations as low as 7.5 μM. Then, we test whether phase separation of nsp8 tetramer could be induced by hairpin RNA with 5′ overhanging. Our result clearly showed that hairpin RNA, reported by Biswal et al.[17], induced phase separation of nsp8 tetramer in a concentration-dependent manner. Thus, our data suggested that the LLPS of nsp8 tetramer depends on both RNA sequence and length. To confirm that the phase-separated droplets were formed by nsp8 binding with RNA, we synthesized 6-FAM labeled R12. Fluorescent microscopy demonstrated that nsp8 and RNA were co-localized in the phase-separated droplet (Fig. 3b).

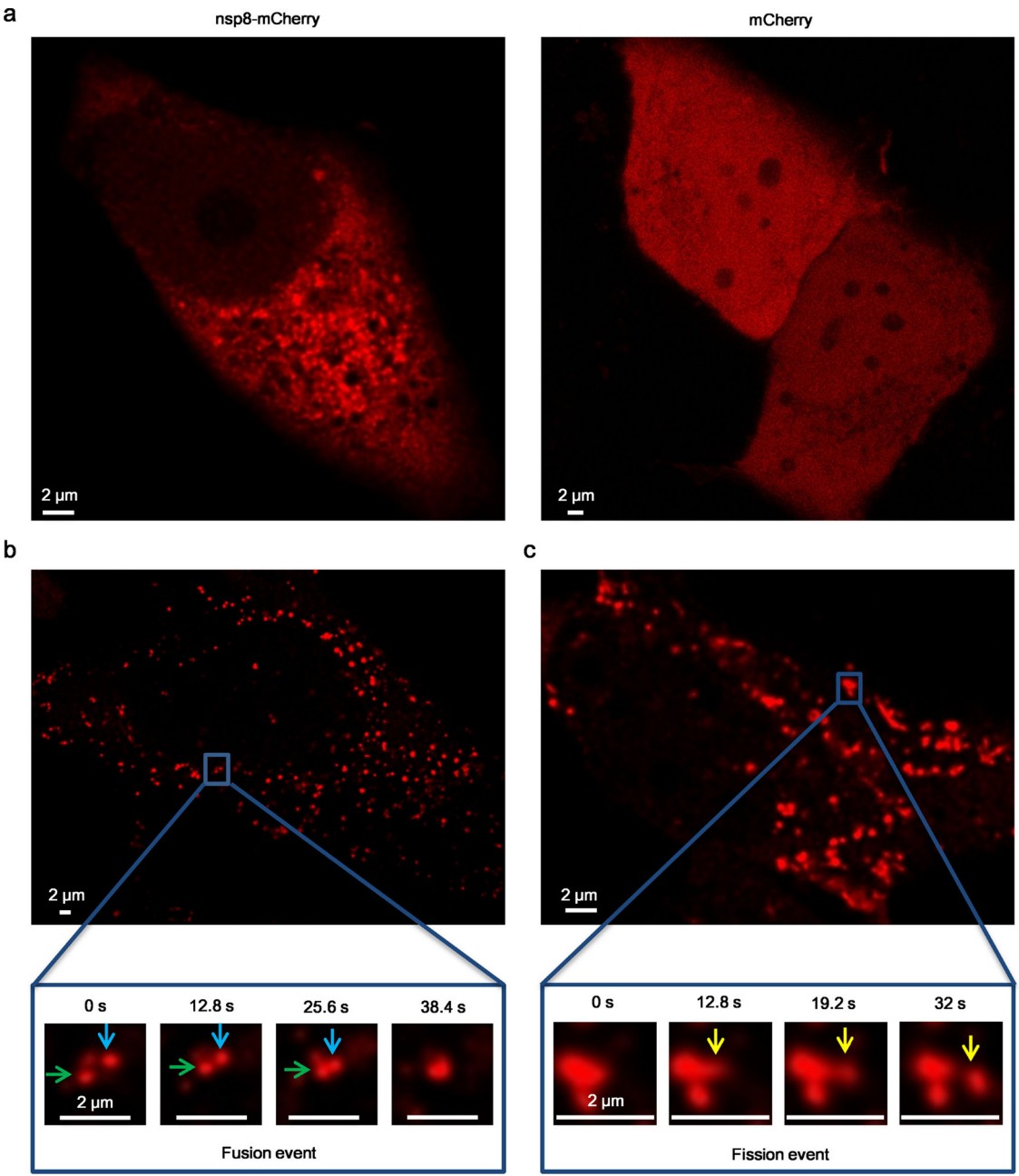

**Fig. 4 LLPS of nsp8 in live cells. a** Representative image of nsp8 formation condensates in Hela cell. Nsp8-mCherry or mCherry was subcloned into pCDNA3.1 and transiently expressed in Hela cells. **b**, **c** LLPS of nsp8 in human bronchial epithelial cells (BEAS-2B). Representative droplet fusion event (**b**) and fission event (**c**) are shown by time course images.

**Nsp8 forms condensates in living cells**. Then, we asked whether nsp8 also undergoes liquid-liquid phase separation in a live human cell line. To address this notion, nsp8 fusion with monomeric red fluorescent protein mCherry at C-terminal (nsp8-mCherry) was over-expressed in Hela cells. By imaging this cell line using fluorescent microscopy, we observed that over-expressed nsp8 could form condensates in the cytoplasm, while over-expressed mCherry was diffuse in the entire cell (Fig. 4a). Then, we test whether nsp8 could also form liquid condensates in cell line relative to SARS-CoV-2 infection. In this experiment, human bronchial epithelial cells (BEAS-2B) were transduced with a lentiviral vector expressing nsp8-mCherry. Confocal fluorescence microscopy showed that nsp8 formed numerous spherical condensates in BEAS-2B cells (Fig. 4b, c). Furthermore, time-lapse observations revealed that nsp8 condensates appeared to be fused or divided rapidly, confirming the liquid-like property of SARS-CoV-2 nsp8 condensates. Taken together, our results demonstrated that SARS-CoV-2 nsp8 undergoes phase separation in solution, as well as in live cells.

## Discussion

Transcription and genome replication of viruses could be substantially amplified by concentrating replication and transcription machinery into a liquid-like structure[11]. In coronavirus, the non-structure protein nsp8 not only is identified as a co-factor for nsp12 mediating RNA elongation, but also exhibits de novo RNA synthesis activity. In this study, we demonstrated that, SARS-

CoV-2 nsp8 can undergo liquid-liquid phase separation in an oligomerization-dependent manner.

Recently, Biswal et al. observed tetramer form of nsp8 through crosslinking assay[17], which is in agreement with our current finding. However, Biswal et al. did not separate nsp8 tetramer from gel filtration analysis. During the purification process, we found that, when the concentration of NaCl in Ni-NTA purification buffer and gel filtration buffer is 300 and 100 mM, respectively, most nsp8 formed oligomers by associating with bacterial nuclear acid, only a small amount of nsp8 could be separated as a dimer, and nsp8 tetramer could not be identified. However, when the concentration of NaCl in the Ni-NTA purification buffer and gel filtration buffer increased to 1000 and 500 mM, respectively, we obtained nsp8 dimer and nsp8 tetramer. Based on our findings, we speculate that, the reason for Biswal et al. not detecting nsp8 tetramer through gel filtration might be that the NaCl concentration in the purification buffer may not be high enough.

Droplet formation of the macromolecule can be modulated by non-covalent modification (e.g., oligomerization state and ligand binding) or physicochemical changes in the environment (e.g., ionic strength)[11]. In this study, we identified the dimer and tetramer forms of nsp8, which undergo phase transformation at a low concentration of NaCl in aqueous solutions. However, they exhibit distinct phase separation behaviors: nsp8 dimers form separated liquid phases depending only on concentrations of the protein and salt, whereas nsp8 tetramers form solid-like aggregates at low salt concentration but can transform into liquid-like droplets with the addition of RNA. The concentration of NaCl is relevant to ionic strength, suggesting phase separation of nsp8 dependent on ionic strength. Thus, our results showed that LLPS of SARS-CoV-2 nsp8 depends on non-covalent modification and physicochemical changes in the environment. Intracellular ionic strength could be altered by cellular signaling events[19]. Further studies are needed to investigate if phase separation of nsp8 can be regulated by cellular signaling events affecting intracellular ionic strength.

More recently, SARS-CoV-2 nucleocapsid (N) protein was reported to undergo LLPS upon binding to RNA or SARS-CoV-2 membrane protein (M)[20–22]. RNA-induced LLPS of N protein is dependent on the length of RNA[23]. Our studies on the phase separation of nsp8 tetramer showed that, its phase separation depends not only on the length of RNA, but also on the sequence of RNA. The physiological relevance of RNA sequence and length-dependent phase separation of nsp8 tetramer remains to be further investigated.

SARS-CoV-2 replication machinery could be concentrated into droplets of N-RNA, suggesting N protein may modulate SARS-CoV-2 replication via LLPS[21]. Here, we determined that nsp8, a component of replication transcription complex (RTC), can form liquid-like droplets in solution and live cells. To our knowledge, nsp8 is the first component of the replication transcription complex in coronavirus that possesses LLPS property. It is interesting to see if nsp8 co-operates with N protein to amplify transcription and replication of SARS-CoV-2 via LLPS. More work is needed to investigate if the primase activity of nsp8 depends on phase separation. Additionally, whether our studies on phase separation of nsp8 can facilitate developing a potential strategy against SARS-CoV-2 remains to be further explored.

The transcription and replication machinery is highly conserved across coronavirus, including Alpha-, Beta-, Gamma-, and Delta-coronavirus[24]. Thus, our results on the phase separation behavior of non-structure protein nsp8 will bring a better understanding of the primer synthesis mechanism and amplified replication of coronavirus, not limited to SARS-CoV-2.

## Methods

**Protein expression and purification.** cDNA fragment encoding SARS-CoV-2 nsp7, nsp8, or truncated nsp8 mutant was cloned into pET21a with C-terminal 8 x His-tag. Plasmids of pET21a-nsp7 and pET21a-nsp8 were provided by Dr. H. Eric Xu. Proteins were expressed in E. coli strain BL21 (DE3) at 16 °C overnight and induced with 0.1 mM IPTG. Bacteria were collected and re-suspended in lysis buffer (20 mM HEPES pH 7.4, 1 M NaCl, 10 mM imidazole, 7 mM β-mercaptoethanol). Cells were lysed by high-pressure homogenization. After centrifugation, the supernatant was loaded onto Ni-NTA resin. After washing with 70 mM imidazole, the bounded protein was eluted with 300 mM imidazole. Proteins were further purified by gel filtration using a HiLoad Superdex 200 (for wild-type nsp8) or 75 (for nsp7 or truncated nsp8) equilibrated against a buffer of 20 mM HEPES pH 7.4, 0.5 M NaCl, 1 mM DTT.

Primers for nsp8ΔN76 mutation: CTTTAAGAAGGAGATATACATATGGAA GATAAACGTGCAAAAG (Forward), and CTTTTGCACGTTTATCTTCCATATG TATA TCTCCTTCTTAAAG (Reverse).

**Small-angle neutron scattering (SANS).** Small-angle neutron scattering measurements were performed at the China spallation neutron source (CSNS). 10 mg/mL nsp8 samples in D2O buffer were measured in quartz cells with a 2 mm optical path length. The scattering experiment times of buffers and samples are 90 min. And the empty beam and empty quartz cell were measured for 15 min for data reduction. Transmissions were measured for 10 min for each sample and the empty beam. All measurements were carried out at 289 K.

The presented SANS data were reduced and corrected for sample transmission, cell scattering, and detector background using reduction algorithms developed based on the Mantid framework[25] provided by the instrument. The neutron scattering intensity has been calibrated to absolute intensity using a secondary standard, Bates poly. The SasView (https://www.sasview.org/) software was employed to calculate the radius of gyration Rg of all samples by Guinier approximation.

**Dynamic light scattering (DLS).** Nsp8 dimer or tetramer with a concentration of 1 mg/mL was dissolved in a buffer containing 20 mM HEPES pH 7.4, and 125 mM or 250 mM or 500 mM NaCl. The DLS analyses were performed using DynaPro NanoStarTM (Wyatt Technologies) at 25 °C. The quartz cells were filled with 100 μL 1 mg/mL nsp8 sample solution for DLS measurement. The samples were measured through a 100 mW He-Ne laser, λ0 = 660 nm, θ = 90°. Ten times successive measurements were performed per sample for average to get the DLS data. The DYNAMICS software (Wyatt Technologies) was employed to analyze the DLS data to get the RH of all samples. Data from three independent experiments were used for analysis.

**Thermo-stability analysis.** SARS-CoV-2 nsp8 dimer and tetramer with concentration 1 mg/mL were dissolved in 20 mM HEPES pH 7.4, and 150 mM or 250 mM or 500 mM NaCl. Thermo-stability analysis was performed with Prometheus NT.48 instrument (Nano Temper Technologies). The scan temperature was increased linearly with a rate of 1 °C per min. Melting temperature (Tm) and onset temperature of aggregation (Tagg) were analyzed with PR.ThermControl software. Data from three independent experiments were used for analysis.

**Imaging of SARS-CoV-2 nsp8 or RNA complex in solution.** For differential interference contrast (DIC) imaging of SARS-CoV-2 nsp8, the proteins were spotted on a glass slide. Imaging was performed with Leica DM4 B.

For fluorescence imaging of SARS-CoV-2 nsp8, the proteins were labeled with a His-tag labeling kit by mixing protein and RED-tris-NTA 2nd Generation dye with a mole ratio of about 600:1 and incubating at room temperature for 15 min. To investigate the effect of RNA on phase separation of SARS-CoV-2 nsp8 tetramer, RNA was added to the RED-tris-NTA labeled protein. ssRNA sequences used in this study, R12: rGrArGrArArUrGrArCrArArA, U12: rUrUrUrUrUrUrUrUrUrUrUrU, U22: rUrUrUrUrUrUrUrUrUrUrUrUrUrUrUrUrUrUrUrUrUrU, hairpin RNA: rUrUrUrU rCrArUrGrCrUrArCrGrCrGrUrArGrUrUrUrUrCrUrArCrGrCrG. For co-localization experiments, R12 was labeled with 6-FAM at 5′ terminal. Before imaging, 10 μL aliquot of sample was placed into a well of glass bottom 384 well plates. Imaging was performed with Zeiss LSM 800 Confocal Laser Scanning Microscopy.

**Fluorescence recovery after photobleaching (FRAP).** Nsp8 dimer phase separation was induced using 1 mg/mL RED-tris-NTA labeled nsp8 dimer in buffer containing 20 mM HEPES pH 7.4, and 50 mM NaCl. FRAP experiments were recorded on a Leica SP8 X STED Laser Scanning Confocal Microscopy using a 100x objective and a 650 nm laser. The region of interest loop (ROI) was bleached with a laser power of 30% and an exposure time of 30 ms. Recovery was imaged at low laser intensity with a power of 10%. Five images and 100 images were obtained before bleach and post-bleach, respectively, with one frame 2 s. Data from three independent experiments were used for analysis.

**Live imaging.** For expressing in Hela cells, nsp8 was fused with mCherry-tag at the C-terminal, and cloned into vector pCDNA3.1. Hela cells (ATCC HTB-22) were

seeded at $2 \times 10^5$ on a glass bottom culture dish (28.2 mm), and cultured in DMEM medium supplemented with 10% FBS. Plasmid pCDNA3.1-nsp8-mCherry (2 μg) was transfected into Hela cells with Lipofectamine 2000 following the manufacturer's instructions.

To prepare nsp8-mCherry stably expressing BEAS-2B cell line, lentivirus vector of pRlenti containing *nsp8-mCherry* and lentiviral packaging plasmids (psPAX2 and pMD2.G) were co-transfected into HEK293T cells (ATCC CRL-1573) in six holes' plate. After 48 h incubation, the virus supernatant was collected by centrifugation at $500 \times g$ for 5 min. For lentiviral infection, BEAS-2B cells (ATCC CRL-9609) were plated with polybrene (8 ug/mL) and 150 uL prepared lentivirus. After 48 h incubation, BEAS-2B cells stably expressing nsp8-mCherry were selected based on antibiotic resistance using 2 μg/mL puromycin. For imaging, BEAS-2B cells stably expressing nsp8-mChery were seeded at $2 \times 10^5$ on a glass bottom culture dish (28.2 mm), and cultured in DMEM medium supplemented with 10% FBS.

Before imaging, the culture medium was replaced with phenol red-free DMEM supplemented with 10% FBS. Imaging was performed with Laser Scanning Confocal Microscopy Leica SP8 X STED.

Primers for pCDNA3.1-nsp8-mCherry
Forward: CCAGCTAGCATGGCCATTGCCAGCG
Reverse: TGGCTCGAGCTGCAGTTTCACTGCAC
Primers for pRlenti-nsp8-mCherry
Forward: CGCGGATCCATGGCCATTGCCAGCGAATTTTC
Reverse: CCGGAATTCCTTGTACAGCTCGTCCATGC

**Statistics and reproducibility**. Dynamic light scattering, thermo-stability analysis, and FRAP experiments were repeated three times. Statistical analyses were performed using GraphPad Prism 7 software.

**Reporting summary**. Further information on research design is available in the Nature Research Reporting Summary linked to this article.

## Data availability
Small-angle neutron scattering (SANS) datasets for nsp8 dimer and tetramer were deposited to Small-Angle Scattering Biological Data Bank (SASBDB) (SASDP56, SASDP66, SASDP76, SASDP86, SASDP96, and SASDPA6). The source data for the graphs presented in the figures of this paper are available in Supplementary Data 1. Uncropped SDS-PAGE gels for Supplementary Fig. 3 and Supplementary Fig. 6 are available in Supplementary Data 2 and 3, respectively.

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

## Acknowledgements
The work was financially supported by the Youth Innovation Promotion Association of the Chinese Academy of Sciences (2018390) to J.Xu., Guangzhou Institute of Respiratory Health Open Project (Funds provided by China Evergrande Group) (2020GIRHHMS08), Guangdong Provincial Key Laboratory of Biocomputing (2016B030301007), Science and Technology Program of Guangzhou (2019B121205010 and 202201010030), Project of The State Key Laboratory of Respiratory Disease (SKLRD-Z-202009 and SKLRD-Z-202213), Guangdong-Hong Kong-Macau Joint Laboratory of Respiratory Infectious Diseases (2019B121205010), and the National Basic Science Data Center "Database of Stem Cells and Metabolic Disease" (NO.NBSDC-DB-16). We gratefully acknowledge the China Spallation Neutron Source (CSNS) for the SANS measurements and the Songshan Lake Deuteration Facility (SLDF) for materials and supplies. We would like to thank the SANS beamline stuff for their support in SANS measurements and the data reduction. We thank Dr. H. Eric Xu for providing the prokaryotic expressing plasmid of the SARS-CoV-2 nsp8 and nsp7.

## Author contributions
J.X., J.L., H.W., and X.J. designed the experiment. J.X., Y.Z., Y.D., and R.C. performed biochemical and cellular experiments. X.J., Y.D., C.M., H.J., T.Z., Y.K., and H.C. collected DLS and SANS data. J.X., J.L., H.W., and X.J. wrote and edited the manuscript.

## Competing interests
The authors declare no competing interests.
