## [Peer Review File · Communications Biology]

Reviewers' comments:

Reviewer #1 (Remarks to the Author):

In their manuscript, Xu et al. report on the phase separation behavior of nsp8, a SARS-CoV-2 non-structural protein. Upon purification of recombinant forms of this protein, the authors report the formation of protein dimer and tetramers. Furthermore, they observe a phase separation-like behavior of the recombinant proteins that is influenced by variations in the buffer NaCl₂ concentration and RNA binding in vitro. Finally, the authors investigate the phase separation behavior of nsp8 when expressed in living cells.

The major question tackled in this paper – the mechanism of de novo RNA synthesis by nsp8 – is of high interest for the current SARS-CoV-2 pandemic and the speculation that phase-separation might play a role in this process is intriguing. In their manuscript, the authors provide first experiments results that might hint towards such a mechanism. However, the presented experiments as well as their description, do not allow to draw conclusions at this point. I would like to recommend several major modification on the manuscript and the experimental procedures in order to sustain the claims raised by the authors.

1. The introduction is very comprehensive and mostly complete. However, in my eyes the authors should specify the research question in the introduction section more clearly. They ask for a mechanism of de novo RNA synthesis by nsp8 but they actually do not directly provide a mechanism. Instead, they follow an interesting research direction related to phase separation, which might impact on the underlying mechanisms. But the phase separation behavior is not the mechanism of RNA synthesis itself. I recommend rewriting of this.
2. Essential experiment information is missing and where experimental procedures are described, the information provided is minimal. The materials and methods section does not provide sufficient information to verify and reproduce the findings. For instance, the methods section does not provide any information on the buffer conditions and measurement parameters used for the DLS measurements. The authors report changes in nsp8 size upon difference in NaCl concentration. Did they make sure that is not due to changes in solvent viscosity? What was the buffer used? I suggest to provide tabular description of the buffer conditions for all figures in a supplementary file.
3. Any information on the constructs and cloning for protein purification and cellular expression is missing.
4. Any information on the microscopy information is missing and no description of the FRAP experiments is included in the manuscript.
5. Similarly, essential information on the data is missing from the figure legends (e.g. description of the graph elements, sampling size). Statistical analysis is missing completely. This does not comply with the NPG requirements and does not adhere to good scientific practice.
6. All quantification (e.g. Figure 1C-F and 2D) should be performed in independent experiments and if possible, with different protein purification batches. Quantitative analysis (e.g. FRAP) should be performed for intracellular nsp8 analysis. Positive and negative controls should be included in the analysis of intracellular nsp8 condensation.
7. Cell experiments: I suggest to performed the experimental observations of phase separation in cells with cell types of physiological relevance to SARS-CoV-2 infection (e.g. bronchial epithelial cells, vascular endothelial cells or T-cells).
8. The authors state that nsp8 forms condensates in living cells and provide only one (!) representative (?) image for this without and intracellular quantification of the phase separation dynamics. Is the phase-separation inducible? How does co-expression of SARS-CoV-2 RNA, like performed in the in vitro experiments, impact on the separation behavior? How do the authors make sure that the structures observed are not simple protein aggregates due to protein overexpression?
9. Could the authors provide any information or experimental description on the existence of intracellular dimer and tetramer?
10. All microscopy images should include scale bars.

In summary, the authors pursue an interesting question that will be appealing to a wider scientific community. However, several additional experimental verification and improvements on data representation should be performed to allow for publication of this manuscript.

Reviewer #2 (Remarks to the Author):

In a report titled "Multiscale characterization reveals oligomerization dependent phase separation of primer-independent RNA polymerase nsp8 from SARS-CoV-2", Xu, Jiang and others describe their observations on the properties and conditions of phase condensates formed by nsp8. The study uniquely reports on this issue. The publication will promote further investigation of this phenomenon and should be published. The study is well done.

The tetramer dimer separation under the conditions studied has not been reported and is also interesting.

The manuscript would be improved if the following major point was dealt with along with some minor points listed further below.

The major point that would be very valuable to address is to investigate whether many of the observed phenomenon are reversible. The authors report that nsp8 separates into dimers and tetramers. If the dimer peak from purification is collected and run on size exclusion again, does the tetramer reappear? Are dimers and tetramers in dynamic equilibrium? Similarly, if the tetramer peak is collected and run on size exclusion again, does the dimer reappear?

The fact that the two fractions behave differently does argue in favor of unique irreversible states. However, there must be something unusual going on if the tetramer cannot break apart and form a dimer. Is there any possibility of a covalent bond (di-sulfide)?

The same is true for the phase condensates. Once they form, are they reversible? Can you get dimer and tetramer again, or, do they form some irreversible aggregate?

The minor issues:

The literature reports that polymerase activity occurs independent of phase condensates. Is there any evidence of polymerase activity in the condensates?

The abstract and conclusion mentions that the phenomenon could facilitate developing novel therapeutics. How does one develop a therapeutic agent targeting this phase behavior? The mechanistic impact is sufficient to justify the paper – not sure the connection to therapeutics is necessary.

Citation 14 mainly discusses a monomer of nsp8 though the size exclusion and multi-angle light scattering do provide evidence for higher oligomerization.

Occasionally a sentence in the manuscript starts with nsp8. The "n" should be capitalized in this instance.

Reviewer #3 (Remarks to the Author):

Xu and colleagues present interesting in vitro data with purified proteins supporting that SARS-CoV-2

NSP8 can form dimers and tetramers that have the potential to condensate and form phase separated entities with potentially different biophysical properties.

Although interesting the manuscript is preliminary and does not explore the functional significance of NSP8 condensates neither does it demonstrate their existence at physiological levels of NSP8 within infected cells.

- The first finding of the manuscript, namely the ability of NSP8 to form dimers and tetramers has been observed by others (Biswal et al., NAR, 2021). It is of note though that there is a difference in the SEC traces presented here compared to those shown in Figure 1B in Biswal et al., where the tetramer cannot be seen. Why is this?

- The findings presented here are not placed within the context of function of the SARS-CoV-2 RdRP. Both NSP8 and NSP7 are required for function of RSP12. How does NSP7 affect the formation of the proposed NSP8 phase separated condensates?

- The authors show that phase separation is dependent on the whole N terminal IDR. Are there any specific/single NSP8 mutations that can disrupt oligomerisation and phase separation?

- It is unclear why the authors use the specific RNA, R12 in figure 3. For example, in Biswal et al., a longer RNA with a 5-prime overhang was used for in vitro functional assays (which would be a welcome addition here). Overall, there is a lot of room here for further exploring the role of RNA in NSP8 phase separation.

- The data presented in Figure 4 is neither of publication quality nor quantity and the Figure has not a proper legend. This is not sufficient to demonstrate that NSP8 can form phase separations in cells, let alone that it does form phase separations in infected cells.

- There is almost a complete absence of Discussion. The authors need to place their findings within the context of the literature. For example, how do the findings presented here relate to work published on SARS-CoV-2 nucleocapsid protein phase separation (Savastano et al., 2020, Nat Comms)? Do other viruses use similar biophysical strategies (reviewed in Lopez et al., 2021, PLoS Path)? What is the benefit for the virus? The authors should also clearly highlight strengths, weaknesses, and when possible, mitigations for the weaknesses of their study.

Minor comments

- Lines 80-82: please add references for the statement about LLPS accelerating reactions in cells.

- Line 90: This manuscript does not include any data on primer synthesis.

- Line 104: Both references 13 and 14 should follow after "previously".

- Lines 168-170: The authors state: "We speculated that RNA binding may induce nsp8 tetramer transition from solid like condensate to liquid liquid phase separation at low NaCl concentration." Why? Is this based on literature?

- As this is a wide readership journal, it would be useful to explain the relevance of changing NaCl concentrations in experiments with purified proteins. How do these relate to cellular environments?

- The language and spelling need a bit of attention throughout.

Reviewer 1

Q: (1) The introduction is very comprehensive and mostly complete. However, in my eyes the authors should specify the research question in the introduction section more clearly. They ask for a mechanism of de novo RNA synthesis by nsp8 but they actually do not directly provide a mechanism. Instead, they follow an interesting research direction related to phase separation, which might impact on the underlying mechanisms. But the phase separation behavior is not the mechanism of RNA synthesis itself. I recommend rewriting of this.

A: We thank the reviewer for this comment. In the revised manuscript, this part was rewritten as following:

“It had been reported that, for a number of viruses, the replication and assembly was taken place within granular structures termed “viral factories”. Recently, viral factory was proposed to be membrane-less organelle driven by liquid-liquid phase separation (LLPS). Concentrating reactive molecules in condensates via liquid-liquid phase separation was considered a powerful mechanism for accelerating biochemical reactions. Proteins tended to LLPS were proposed to exhibit features including intrinsically disordered, modularity, nucleic acid binding and oligomeric nature, et.al. Sequence analysis showed that SARS-CoV-2 nsp8 contains potential IDR at the N terminus (Fig. S1), which is critical for RNA binding, and contains conserved catalytic D/ExD/E motif as characterized in SARS-CoV nsp8. Structural analysis also showed that the free N terminus is flexible in several reported structures, implying that nsp8 may undergo phase separation. Here, we investigated LLPS of nsp8 of SARS CoV-2, unveiling oligomerization dependent phase separation behavior of nsp8.”

Q: (2) Essential experiment information is missing and where experimental procedures are described, the information provided is minimal. The materials and methods section does not provide sufficient information to verify and reproduce the findings. For instance, the methods section does not provide any information on the buffer conditions and measurement parameters used for the DLS measurements. The authors report changes in nsp8 size upon difference in NaCl concentration. Did they make sure that is not due to changes in solvent viscosity? What was the buffer used? I suggest to provide tabular description of the buffer conditions for all figures in a supplementary file.

A: We thank the reviewer for the comment. Information on the buffer conditions has been provided in method section and Figure legend, as following: “Nsp8 dimer or tetramer with concentration of 1 mg/mL was dissolved in buffer containing 20 mM HEPES pH 7.4, and 125 mM or 250 mM or 500 mM NaCl.”

With regard to the solvent viscosity, the relationship between viscosity of solution and hydrodynamic radius follows the Stokes-Einstein equation.

$$D = \frac{k_B T}{6\pi\eta R_h}$$

, in which D is the diffusion coefficient, k_B is the Boltzmann constant, T is the temperature, η is the dynamic viscosity of solution (Pa.s), and R_h is the hydrodynamic radius of particle.

The NaCl concentration has little effect on the dynamic viscosity of solution in 0 M to 0.5 M (Journal of Physical & Chemical Reference Data, 1981, 10(1):71-88). The dynamic viscosity of 0 M NaCl solution is 1.00 Pa.s, while the 0.5 M NaCl solution is 1.04 Pa.s.

Q: (3) Any information on the constructs and cloning for protein purification and cellular expression is missing.

A: We thank the reviewer for the comment. Information on the constructs and cloning for protein purification and cellular expression was provided in the revised method section.

For purification, “cDNA fragment encoding SARS-CoV-2 nsp7, nsp8 or truncated nsp8 mutant was cloned into pET21a with C-terminal 8 x His tag.”

“For expression in Hela cells, nsp8 was fused with mCherry-tag at C terminal, and cloned into vector pCDNA3.1.”

“To prepare nsp8-mChery stably expressing BEAS-2B cell line, lentivirus vector of pKD containing nsp8-mChery and lentiviral packaging plasmids (psPAX2 and pMD2.G) were co-transfected into HEK293T cells in six holes' plate. After 48 h incubation, the virus supernatant was collected by centrifugation at 500g for 5min. For lentiviral infection, BEAS2B cells were plated with polybrene (8 ug/mL) and 150 uL prepared lentivirus. After 48 h incubation, BEAS-2B cells stably expressing nsp8-mChery were selected based on antibiotic-resistance using 2 μ g/mL puromycin.”

Q: (4) Any information on the microscopy information is missing and no description of the FRAP experiments is included in the manuscript.

A: We thank reviewer for the comment. Information of the FRAP experiments was provided in method section as following: “Nsp8 dimer phase separation was induced using 1 mg/mL RED-tris-NTA labeled nsp8 dimer in buffer containing 20 mM HEPES pH 7.4, and 50 mM NaCl. FRAP experiments were recorded on a Leica SP8 X STED Laser Scanning Confocal Microscopy using a 100x objective and a 650 nm laser. Region of interest loop (ROI) was bleached with laser power of 30% and exposure time 30 ms. Recovery was imaged at low laser intensity with power of 10%. 5 images and 100 images were obtained before bleach and post-bleach, respectively, with one frame 2 s.”

Q: (5) Similarly, essential information on the data is missing from the figure legends (e.g. description of the graph elements, sampling size). Statistical analysis is missing completely. This does not comply with the NPG requirements and does not adhere to good scientific practice.

A: We thank the reviewer for this comment. In the revision, sampling size and error bar was provided in the legend of Figure 1 and Figure 2. For DLS experiment, thermo-stability analysis and FRAP experiment, data were obtained from three independent experiment.

Q: (6) All quantification (e.g. Figure 1C-F and 2D) should be performed in independent experiments and if possible, with different protein purification batches. Quantitative analysis (e.g. FRAP) should be performed for intracellular nsp8 analysis. Positive and negative controls should be included in the analysis of intracellular nsp8 condensation.

A: We thank the reviewer for this comment. For DLS experiment, thermo-stability analysis and FRAP experiment, data were obtained from three independent experiment.

For intracellular nsp8 analysis, we failed to observe rapid recovery of fluorescence after photobleaching, due to small size of nsp8 condensates in cell. However, time-lapse observations revealed that nsp8 condensates in bronchial epithelial cell appear to be fused or divided rapidly (Fig. 4B and C), indicating liquid-like property of SARS-CoV-2 nsp8 condensates in living cells.

Figure 4 (B and C) LLPS of nsp8 in human bronchial epithelial cells (BEAS-2B). Representative droplet fusion event (B) and fission event (C) are shown by time course images.

Q: (7) Cell experiments: I suggest to performed the experimental observations of phase separation in cells with cell types of physiological relevance to SARS-CoV-2 infection (e.g. bronchial epithelial cells, vascular endothelial cells or T-cells).

A: We thank the reviewer for this comment. We performed the experimental observations of nsp8 phase separation in bronchial epithelial cells (BEAS-2B). As shown in Figure 4B and C, liquid like condensates can be observed in BEAS-2B cells.

Q: (8) The authors state that nsp8 forms condensates in living cells and provide only one (!) representative (?) image for this without and intracellular quantification of the phase separation dynamics. Is the phase-separation inducible? How does co-expression of SARS-CoV-2 RNA, like performed in the in vitro experiments, impact on the separation behavior? How do the authors make sure that the structures observed are not simple protein aggregates due to protein overexpression?

A: We thank the reviewer for this comment. The representative image, for nsp8 forming condensates in live Hela cells, was presented in Figure 4A. Nsp8 condensates were also observed

in live bronchial epithelial cells (BEAS-2B) (Figure 4B and C). Fusion and fission events can be observed in live BEAS-2B, indicating nsp8 condensates are liquid-like, not simple protein aggregates.

In solution, phase separation of nsp8 dimer can be induced by decreasing NaCl concentration, and phase separation of nsp8 dimer can be induced by decreasing NaCl concentration and additional RNA. For live cells, LLPS of nsp8 can be observed without additional treatment. Numerous RNAs were distributed into whole cell, which may have impact on the phase separation of nsp8.

Q: (9) Could the authors provide any information or experimental description on the existence of intracellular dimer and tetramer?

A: We thank the reviewer for this comment. Nsp8 can form condensate in cells. However, it is very difficult to characterize the precise oligomer state of intracellular nsp8.

Q: (10) All microscopy images should include scale bars.

A: We thank the reviewer for this comment. Scale bars were added on all microscopy images in the revision.

Reviewer 2

Q: (1) The authors report that nsp8 separates into dimers and tetramers. If the dimer peak from purification is collected and run on size exclusion again, does the tetramer reappear? Are dimers and tetramers in dynamic equilibrium? Similarly, if the tetramer peak is collected and run on size exclusion again, does the dimer reappear?

A: We thank the reviewer for the suggestions. To test whether the dimer and tetramer of nsp8 are in dynamic equilibrium, here, dimer or tetramer of nsp8 eluted from gel filtration was collected and analyzed by gel filtration again after standing for 10 days. Gel filtration analysis showed that oligomer state changes only can be observed for small portion of dimer or tetramer (Fig S2), suggesting oligomer state of nsp8 is rather stable in solution.

Figure S2 Dimer (A) and tetramer (B) form nsp8 were analyzed with Superdex Increase 200, respectively, after standing for 10 days. Before loading sample, Superdex Increase 200 was equilibrated with buffer containing 20 mM HEPES pH 7.4, 0.5 M NaCl, 1 mM DTT.

Q: (2) The fact that the two fractions behave differently does argue in favor of unique irreversible states. However, there must be something unusual going on if the tetramer cannot break apart and form a dimer. Is there any possibility of a covalent bond (di-sulfide)?

A: We thank the reviewer for this comment. SDS-PAGE result showed that dimer form and tetramer form nsp8 almost completely turned to monomer by SDS (Fig S3), even without reducing agent, indicating oligomerization of nsp8 is disulfide bond independent.

Figure S3 nsp8 dimer or tetramer at concentration of 0.5 mg/ml dissolved in buffer 20 mM HEPES pH 7.4, 0.5 M NaCl, was mixed with loading buffer with or without 0.7 M β -mercaptoethanol (β -ME) and analyzed by SDS-PAGE.

Q: (3) The same is true for the phase condensates. Once they form, are they reversible? Can you get dimer and tetramer again, or, do they form some irreversible aggregate?

A: We thank the reviewer for this comment. As shown in Figure 2C and E, microscopy observation showed that phase condensates of nsp8 dimer or tetramer, can be reversed upon increasing NaCl concentration. Upon increasing concentration to 275 mM, no significant aggregation can be observed under microscopy.

Minor comments

Q: (1) The literature reports that polymerase activity occurs independent of phase condensates. Is there any evidence of polymerase activity in the condensates?

A: We thank the reviewer for this comment. It was reported that gene expression rates would be amplified by forming membrane-less organelles through LLPS (Wei, M.T. et al., Nature cell biology 22, 1187-1196 (2020)). It was also proposed that genome replication of virus often take place within “viral factories” with dynamic properties driven by LLPS (Novoa, R.R. et al., Biol Cell 97, 147-172 (2005); Lopez et al., Plos Pathog 17 (2021)).

Q: (2) The abstract and conclusion mentions that the phenomenon could facilitate developing novel therapeutics. How does one develop a therapeutic agent targeting this phase behavior? The mechanistic impact is sufficient to justify the paper – not sure the connection to therapeutics is necessary.

A: We thank the reviewer for this comment. In the revised manuscript, we have removed the sentence about developing novel therapeutics in abstract and conclusion section.

Q: (3) Citation 14 mainly discusses a monomer of nsp8 though the size exclusion and multi-angle light scattering do provide evidence for higher oligomerization.

A: Thanks the reviewer for this comment. In the revised manuscript, this reference was replaced by Biswal et al. Nucleic Acids Res 49, 5956-5966 (2021).

Q: (4) Occasionally a sentence in the manuscript starts with nsp8. The "n" should be capitalized in this instance.

A: We are apologizing for this mistake. We corrected this mistake in the revised manuscript.

Reviewer 3

Q: (1) The first finding of the manuscript, namely the ability of NSP8 to form dimers and tetramers has been observed by others (Biswal et al., NAR, 2021). It is of note though that there is a difference in the SEC traces presented here compared to those shown in Figure 1B in Biswal et al., where the tetramer cannot be seen. Why is this?

A: Thanks the reviewer for this comment. Indeed, Biswal et al observed tetramer form nsp8 through crosslinking assay, but not by gel filtration analysis (Biswal et al. Nucleic Acids Res 49, 5956-5966 (2021)).

During the purification process, we found that, when concentration of NaCl in Ni-NTA purification buffer and gel filtration buffer is 300 mM and 100 mM, respectively, most of nsp8 forming multi-oligomer by association with bacterial nuclear acid, only small amount of nsp8 can be separated as dimer, and nsp8 tetramer cannot be identified. However, when concentration of NaCl in Ni-NTA purification buffer and gel filtration buffer increased to 1000 mM and 500 mM, we can obtain nsp8 dimer and nsp8 tetramer. For the reason why Biswal et al cannot identify tetramer, we think that the concentration of NaCl in their purification process may not be high enough to separate nsp8 tetramer through gel filtration

Q: (2) The findings presented here are not placed within the context of function of the SARS-CoV-2 RdRP. Both NSP8 and NSP7 are required for function of RSP12. How does NSP7 affect the formation of the proposed NSP8 phase separated condensates?

A: We thank the reviewer for this comment. In the revised manuscript, we test if phase separation of nsp8 can be enhanced or disrupted by nsp7. As shown in Fig S5, by mixing with nsp7, phase transformation behavior of nsp8 dimer or nsp8 tetramer did not exhibit significant changes.

Figure S5 Phase transformation behavior of nsp8 can't be affected by mixing with nsp7. (A) Differential interference contrast (DIC) imaging of nsp8 dimer (2 mg/mL), mixture of nsp8 dimer (2 mg/mL) and nsp7 (1 mg/mL), and nsp7 (1 mg/mL), dissolved in buffer 20 mM HEPES pH 7.4, and 100 mM NaCl. (B) Differential interference contrast (DIC) imaging of nsp8 tetramer (1 mg/mL), mixture of nsp8 tetramer (1 mg/mL) and nsp7 (0.5 mg/mL), and nsp7 (0.5 mg/mL), dissolved in buffer 20 mM HEPES pH 7.4, and 100 mM NaCl.

Q: (3) The authors show that phase separation is dependent on the whole N terminal IDR. Are there any specific/single NSP8 mutations that can disrupt oligomerisation and phase separation?

A: We thank the reviewer for this comment. As shown in Figure 2F, phase separation of nsp8 dimer was partially inhibited, but not totally abolished by deletion N terminal IDR. And, without the N-terminal IDR, tetramer of nsp8 C-terminal domain also forms condensates or aggregation at 50 mM NaCl (Figure 2G). These results suggest that, both of N terminal IDR and C terminal domain are involved in phase transformation of nsp8. Taken together, it is very difficult to disrupt oligomerisation and phase separation of nsp8, simply by specific/single NSP8 mutations.

Q: (4) It is unclear why the authors use the specific RNA, R12 in figure 3. For example, in Biswal et al., a longer RNA with a 5-prime overhang was used for in vitro functional assays (which would be a welcome addition here). Overall, there is a lot of room here for further exploring the role of RNA in NSP8 phase separation.

A: We thank the reviewer for this suggestion. In the revised manuscript, we test whether phase separation of nsp8 tetramer could be induced by hairpin RNA with 5' overhanging as reported by Biswal et al. Our result clearly showed that, the hairpin RNA could induce phase separation of nsp8 tetramer in concentration dependent manner.

Before imaging, nsp8 tetramer at 0.25 mg/mL was labeled with RED-tris-NTA and mixed with hairpin RNA in buffer of 20 mM HEPES pH 7.4, 50 mM NaCl

Q: (5) The data presented in Figure 4 is neither of publication quality nor quantity and the Figure has not a proper legend. This is not sufficient to demonstrate that NSP8 can form phase separations in cells, let alone that it does form phase separations in infected cells.

A: We thank the reviewer for this comment. In the revised manuscript, we redo the Figure 4 and Legend. The image for phase separation in Hela was replaced by a representative image with higher quality. Moreover, nsp8 condensates also can be observed in bronchial epithelial cell. The dynamic property of nsp8 condensate was confirmed by fusion or fission events.

Figure 4. LLPS of nsp8 in live cells. (A) Representative image of nsp8 formation condensates in HeLa cell. Nsp8-mCherry or mCherry was subcloned into pCDNA3.1 and transiently expressed in HeLa cells. (B and C) LLPS of nsp8 in human bronchial epithelial cells (BEAS-2B). Representative droplet fusion event (B) and fission event (C) are shown by time course images.

Q: (6) There is almost a complete absence of Discussion. The authors need to place their findings within the context of the literature. For example, how do the findings presented here relate to work published on SARS-CoV-2 nucleocapsid protein phase separation (Savastano et al., 2020, Nat Comms)? Do other viruses use similar biophysical strategies (reviewed in Lopez et al., 2021,

PLoS Path)? What is the benefit for the virus? The authors should also clearly highlight strengths, weaknesses, and when possible, mitigations for the weaknesses of their study.

A: Thanks the reviewer for this comment. In the revised manuscript, discussion was added as following: “Droplet formation of macromolecule can be modulated by non-covalent modification (e.g., oligomerization state and ligand binding), or physicochemical changes in environment (e.g., ionic strength). In this study, we identified the dimer and tetramer forms of nsp8, which undergo phase transformation at low concentration of NaCl in aqueous solutions. However, they exhibit distinct phase separation behaviors: nsp8 dimers form separated liquid phases depending only on concentrations of the protein and salt, whereas nsp8 tetramers form solid-like aggregates at low salt concentration but can transform to liquid-like droplets with the addition of RNA. Concentration of NaCl is relevant to ionic strength, suggesting phase separation of nsp8 dependent on ionic strength. Thus, our results showed that LLPS of SARS-CoV-2 nsp8 depends on non-covalent modification and physicochemical changes in environment. Intracellular ionic strength could be altered by cellular signaling events. Further studies are needed to investigate if phase separation of nsp8 can be regulated by cellular signaling events affecting intracellular ionic strength.

More recently, SARS CoV-2 nucleocapsid (N) protein was reported to undergo LLPS upon binding to RNA or SARS CoV-2 membrane protein (M). In addition, SARS-CoV-2 replication machinery could be concentrated into droplet of N-RNA, suggesting N protein may modulate SARS-CoV-2 replication via LLPS. Here, we determined that nsp8, a component of replication transcription complex (RTC), can form liquid-like droplet in solution and live cells. To our knowledge, nsp8 is the first component of the replication transcription complex in coronavirus that possesses LLPS property. It is interesting to see if nsp8 co-operates with N protein to amplify transcription and replication of SARS-CoV-2 via LLPS. Additionally, more work is needed to investigate if primase activity of nsp8 depends on phase separation.

The transcription and replication machinery is highly conserved across coronavirus, including Alpha-, Beta-, Gamma-, and Delta-coronavirus. Thus, our results on the phase separation behavior of non-structure protein nsp8 will bring better understanding on the primer synthesis mechanism and amplified replication of coronavirus, not limited to SARS-CoV-2.”

Minor comments

Q: (1) Lines 80-82: please add references for the statement about LLPS accelerating reactions in cells.

A: Thanks the reviewer for this comment. In revised manuscript, references (Lopez, N. et al. *Plos Pathog* 17 (2021); Zheng, C.H. et al. *Front Physiol* 12 (2021); Wei, M.T. et al., *Nature cell biology* 22, 1187-1196 (2020)) for this statement were added.

Q: (2) Line 90: This manuscript does not include any data on primer synthesis.

A: Thanks the reviewer for this comment. In the revised manuscript, this sentence was replaced by “Here, we investigated LLPS of nsp8 of SARS CoV-2, unveiling oligomerization dependent phase separation behavior of nsp8”

Q: (3) Line 104: Both references 13 and 14 should follow after “previously”.

A: We are apologizing for this mistake. In the revised manuscript, the reference (Biswal et al. *Nucleic Acids Res* 49, 5956-5966 (2021)) was added.

Q: (4) Lines 168-170: The authors state: “We speculated that RNA binding may induce nsp8 tetramer transition from solid like condensate to liquid liquid phase separation at low NaCl concentration.” Why? Is this based on literature?

A: Thanks the reviewer for this comment. It was proposed that droplet property of protein might be altered after binding to RNA (Zheng, C.H. et al. *Front Physiol* 12 (2021))

Q: (5) As this is a wide readership journal, it would be useful to explain the relevance of changing NaCl concentrations in experiments with purified proteins. How do these relate to cellular environments?

A: Thanks the reviewer for this comment. In the revised manuscript, we discussed this in discussion section as: “Concentration of NaCl is relevant to ionic strength, suggesting phase separation of nsp8 dependent on ionic strength. Thus, our results showed that LLPS of SARS-CoV-2 nsp8 depends on non-covalent modification and physicochemical changes in environment. Intracellular ionic strength could be altered by cellular signaling events. Further

studies are needed to investigate if phase separation of nsp8 can be regulated by cellular signaling events affecting intracellular ionic strength.”

Q: (6) The language and spelling need a bit of attention throughout.

A: Thanks the reviewer for the comment. We have carefully proofread and corrected spelling mistakes.

Reviewers' comments:

Reviewer #1 (Remarks to the Author):

The authors have performed several new experiments that answer my concerns raised in the first revision round. However, the statistical analysis is still missing. I consider the information now added to the methods section as bare minimum required for reproduction of the experiments. I can recommend publication of the revised article with minor revisions.

Reviewer #2 (Remarks to the Author):

This article has been thoroughly reviewed. The authors have done their best to address the issues raised by reviewers. There remain some areas that need to be fully investigated. The multimeric state of Nsp8 is enigmatic and several features have been ascribed to various multimers. The observed phase transitions need to be further investigated and this article is an important step in identifying them and characterizing them.

Reviewer #3 (Remarks to the Author):

I would like to thank the authors for addressing some of my comments. There are still some points that I believe are essential to be addressed before this is acceptable for publication.

- The response to my first question (It is of note though that there is a difference in the SEC traces presented here compared to those shown in Figure 1B in Biswal et al., where the tetramer cannot be seen. Why is this?) is interesting and should be added to the Discussion.
- The authors did not include a section with limitations of their study in their Discussion. This is absolutely necessary.
- In Figure 3, the image is representative of what (how many experiments, how many fields of view)? Also, the effects of hairpin RNA occur at much lower concentrations than the U12 RNA, why is this? The authors should discuss. Also, in Figure 3A, are the images for U12 75 microM and hairpin RNA 2.5 microM indicative of phase separation or protein aggregation? This should be discussed in the manuscript. Overall, I still think that the role of RNA in NSP8 phase separation is not sufficiently explored or at least discussed.

Reviewer 1

Q: However, the statistical analysis is still missing. I consider the information now added to the methods section as bare minimum required for reproduction of the experiments.

A: We thank the reviewer for this comment. DLS experiment, thermo-stability analysis and FRAP experiment were repeated at least three times. “Data from three independent experiments were used for analysis” was added to the methods section for DLS experiment, thermo-stability analysis and FRAP experiment.

Reviewer 2

Q: The authors have done their best to address the issues raised by reviewers. There remain some areas that need to be fully investigated. The multimeric state of Nsp8 is enigmatic and several features have been ascribed to various multimers. The observed phase transitions need to be further investigated and this article is an important step in identifying them and characterizing them.

A: We thank the reviewer for the positive comments. We believe that our current results provide an important step for further study on the functional implications of nsp8 LLPS.

Reviewer 3

Q: (1) The response to my first question (It is of note though that there is a difference in the SEC traces presented here compared to those shown in Figure 1B in Biswal et al., where the tetramer cannot be seen. Why is this?) is interesting and should be added to the Discussion.

A: Thanks the reviewer for the comment. We have added a paragraph for the difference in SEC traces between our work and the published results in the discussion section as following:

“Recently, Biswal et al. observed tetramer form of nsp8 through crosslinking assay¹⁷, which is in agreement with our current finding. However, Biswal et al. did not separate nsp8 tetramer from gel filtration analysis. During the purification process, we found that, when concentration of NaCl in Ni-NTA purification buffer and gel filtration buffer is 300 mM and 100 mM, respectively, most nsp8 formed oligomers by associating with bacterial nuclear acid, only a small amount of nsp8 could be separated as dimer, and nsp8 tetramer could not be identified. However, when concentration of NaCl in Ni-NTA purification buffer and gel filtration buffer increased to 1000

mM and 500 mM, respectively, we obtained nsp8 dimer and nsp8 tetramer. Based on our finding, we speculate that, the reason for Biswal et al. not detecting nsp8 tetramer through gel filtration might be that the NaCl concentration of in purification buffer may not be high enough.”

Q: (2) The authors did not include a section with limitations of their study in their Discussion. This is absolutely necessary.

A: Thanks the reviewer for the comment. In the discussion section, the limitations for our study have been presented as the following sentences:

“Further studies are needed to investigate if phase separation of nsp8 can be regulated by cellular signaling events affecting intracellular ionic strength”

“The physiological relevant of RNA sequence and length dependent phase separation of nsp8 tetramer remains to be further investigated.”

“It is interesting to see if nsp8 co-operates with N protein to amplify transcription and replication of SARS-CoV-2 via LLPS. More work is needed to investigate if primase activity of nsp8 depends on phase separation. Additionally, whether our studies on phase separation of nsp8 can facilitate developing novel strategy against SARS-CoV-2 remains to be further investigated.”

Q: (3) In Figure 3, the image is representative of what (how many experiments, how many fields of view)? Also, the effects of hairpin RNA occur at much lower concentrations than the U12 RNA, why is this? The authors should discuss. Also, in Figure 3A, are the images for U12 75 microM and hairpin RNA 2.5 microM indicative of phase separation or protein aggregation? This should be discussed in the manuscript. Overall, I still think that the role of RNA in NSP8 phase separation is not sufficiently explored or at least discussed.

A: Thanks the reviewer for the comment. In Figure 3, the image is representative of at least three independent experiments or three fields of view.

As we concluded in results section that, nsp8 tetramer forms solid like aggregation with 75 μ M U12, while forms phase separation with 75 μ M R12, 7.5 μ M U12 or 2.5 μ M hairpin RNA, indicating LLPS of nsp8 tetramer dependent on both RNA sequence and length.

In discussion section, we further discussed these results with: “More recently, SARS CoV-2

nucleocapsid (N) protein was reported to undergo LLPS upon binding to RNA or SARS CoV-2 membrane protein (M).” And then we added: “RNA induced LLPS of N protein is dependent on length of RNA. Our studies on phase separation of nsp8 tetramer showed that, its phase separation is not only dependent on the length of RNA, but also dependent on the sequence of RNA. The physiological relevant of RNA sequence and length dependent phase separation of nsp8 tetramer remains to be further investigated.”

REVIEWERS' COMMENTS:

Reviewer #2 (Remarks to the Author):

The Authors have worked really hard to address reviewer issues. Given that work, it must be very frustrating to them to take so long to publish. The science now available in this manuscript is of much higher standard than their first draft. The data should be made available to the community as soon as possible.

Reviewer #3 (Remarks to the Author):

Thank you for addressing my final comments. This is now acceptable for publication.